# The Effect of an Attenuated Live Vaccine against Salmonid Rickettsial Septicemia in Atlantic Salmon (*Salmo salar*) Is Highly Dependent on Water Temperature during Immunization

**DOI:** 10.3390/vaccines12040416

**Published:** 2024-04-15

**Authors:** Rolf Hetlelid Olsen, Frode Finne-Fridell, Marianne Bordevik, Anja Nygaard, Binoy Rajan, Marius Karlsen

**Affiliations:** 1PHARMAQ AS, 0275 Oslo, Norway; rolf.hetlelid-olsen@zoetis.com (R.H.O.); frode.finne-fridell@puresalmontech.com (F.F.-F.); marianne.bordevik@zoetis.com (M.B.); anja.nygaard@zoetis.com (A.N.); binoy.rajan@zoetis.com (B.R.); 2Pure Salmon Technology, 3241 Sandefjord, Norway

**Keywords:** Atlantic salmon, *Piscirickettsia salmonis*, salmonid rickettsial septicemia (SRS), vaccine, ALPHA JECT LiVac SRS

## Abstract

Salmonid Rickettsial Septicemia (SRS), caused by the bacterium *Piscirickettsia salmonis*, is the main reason for antibiotic usage in the Chilean aquaculture industry. In 2016, a live attenuated vaccine (ALPHA JECT LiVac^®^ SRS, PHARMAQ AS) was licensed in Chile and has been widely used in farmed salmonids since then. In experimental injection and cohabitation laboratory challenge models, we found that the vaccine is effective in protecting Atlantic salmon (*Salmo salar*) for at least 15 months against *P. salmonis*-induced mortality. However, the protection offered by the vaccine is sensitive to temperature during immunization. Fish vaccinated and immunized at 10 °C and above were well protected, but those immunized at 7 °C and 8 °C (the lower end of the temperature range commonly found in Chile) experienced a significant loss of protection. This temperature-dependent loss of effect correlated with the amount of vaccine-strain RNA detected in the liver the first week after vaccination and with *in vitro* growth curves, which failed to detect any growth at 8 °C. We found that good vaccine efficacy can be restored by exposing fish to 15 °C for the first five days after vaccination before lowering the temperature to 7 °C for the remaining immunization period. This suggests that maintaining the correct temperature during the first few days after vaccination is crucial for achieving a protective immune response with ALPHA JECT LiVac^®^ SRS. Our results emphasize the importance of temperature control when vaccinating poikilothermic animals with live vaccines.

## 1. Introduction

Salmonid Rickettsial Septicemia (SRS), caused by the facultative intracellular bacterium *Piscirickettsia salmonis*, is one of the main disease problems facing Chilean salmonid aquaculture, alongside infestation with sea lice (*Caligus rogercresseyi*) [1]. SRS was first identified in the late 1980s in the Puerto Montt area, where cases of unusually high mortality rates, reaching up to 90%, were reported in Coho salmon (*Oncorhynchus kisutch*). Diseased fish showed symptoms such as lowered hematocrit, swollen kidneys, enlarged spleen, and, in some cases, a mottled liver [2]. A Rickettsia-like bacterium was successfully isolated from affected tissues and identified as *P. salmonis* [3]. Since then, it has become evident that the bacterium can infect a broad range of fish species, including Atlantic salmon (*Salmo salar*) [4,5].

The clinical outcome of a *P. salmonis* infection is relatively well controlled in Chile by the use of antibiotics [6]. While this ensures that the high mortalities observed in earlier years no longer occur, it also results in significantly higher antibiotic use in Chilean aquaculture compared to other salmon-producing countries. Vaccination against *P. salmonis* is possible, and numerous vaccines based on inactivated whole-cell bacterins in water-in-oil emulsions have been available on the market since 1999 [7]. In 2016, a live-attenuated vaccine, ALPHA JECT LiVac^®^ SRS (referred to as “AJ LiVac SRS”, PHARMAQ AS), was licensed and quickly adopted by the industry. Around 95% of Atlantic salmon in Chilean regions X and XI (stocked with 141 million individuals) were vaccinated with AJ LiVac SRS in 2022. A decline in Chilean antibiotic use was observed in the years after the introduction of the vaccine, but in recent years, the use of antibiotics has started to rise again [8]. This has led to some controversy regarding the efficacy of the vaccine, and Chilean researchers recently reported that they found no effect of vaccination with AJ LiVac SRS in a laboratory challenge of vaccinated fish [9].

Live vaccines can, however, be sensitive to errors in their use, particularly so in poikilothermic animals, since the vaccine strain usually needs to replicate to some extent in the vaccinated animal to produce an immune response. Failure to provide the correct conditions for propagation in the host may result in a severe loss of immunity in the vaccinated animal [10]. *P. salmonis* growth is significantly affected by the range of temperatures commonly found in the Chilean farming industry [3,11]. We have therefore assessed the effect of AJ LiVac SRS under different temperature conditions and challenge models. We find that the vaccine is indeed highly effective in reducing mortality due to *P. salmonis* in these models, but the temperature during the first five days after vaccination is crucial for obtaining good protection, likely due to little or no bacterial propagation at 8 °C or below.

These data provide insight into the importance of temperature on *in vivo* vaccine-strain propagation the days after vaccination. Hopefully, this can reduce sub-optimal use of AJ LiVac SRS and perhaps also serve as an example of this general principle that is of relevance on a broader scale for all live vaccines used under ambient temperature conditions.

## 2. Materials and Methods

### 2.1. Animal Trials, Study Sites, and Ethics

A total of four experimental vaccination and challenge trials in Atlantic salmon (*Salmo salar*) were conducted in the period 2015–2019 at VESO Vikan and Industrilaboratoriet i Bergen (ILAB). The trials had approvals from the Norwegian Food Safety Authority under license IDs 6832 and 7532.

### 2.2. Vaccines and Vaccination

The monovalent, live attenuated SRS vaccine ALPHA JECT LiVac^®^ SRS (PHARMAQ AS, Overhalla, Norway) and the multivalent water-in-oil based vaccine ALPHA JECT 5-1 (PHARMAQ AS) were used in the studies. ALPHA JECT^®^ 5-1 (AJ 5-1) is a pentavalent vaccine that contains inactivated *P. salmonis*, *Vibrio ordalii*, *Aeromonas salmonicida* subspecies *salmonicida*, infectious salmon anemia virus (ISAV), and infectious pancreatic necrosis virus (IPNV). Both vaccines were administered by intraperitoneal injection, with a dose of 0.1 mL per fish. The injection was administered using Socorex vaccination guns, and the dose was tested and calibrated by injecting 3 doses into a pre-calibrated disposable syringe. Thawing and dilution of the AJ LiVac SRS vaccine were performed according to the recommendations given in the package SEr insert, except the frozen vaccine was kept at −80 °C instead of in the vapor phase of liquid nitrogen (ca. −135 °C). The thawing method consisted of the vaccine vial being taken out of −80 °C and put into a water bath kept at 25 °C for approximately 3 min until all ice had melted. Immediately after thawing, the vial was diluted in a 1000 mL container of sodium chloride solution (Ecoflac^®^ plus 0.9% NaCl, B. Braun, Melsungen, Germany) using a transfer cap (Ecoflac^®^ Connect, B. Braun). The dilution was mixed thoroughly before use.

### 2.3. Challenge Material and Challenge

Following the period of immunization, the water temperature was elevated to 15 °C by an incremental 2 °C daily temperature increase. The fish were starved for 24 h, anesthetized using Tricain (80 mg/L, PHARMAQ AS), and injected intraperitoneally with 0.1 mL of a challenge inoculum containing 6 × 10^3^ tissue culture infective dose _50_ (TCID_50_) of *P. salmonis* isolate AL10007, a Chilean isolate belonging to the EM90 genogroup. For the tank challenged by cohabitation, a gradual elevation of temperature to 17 °C and, eventually, 19 °C (60 d.p.c.) was started at 40 d.p.c. to stress the groups. Mortalities were then recorded on a daily basis until the termination of the experiment.

### 2.4. Trial 1—Vaccine Efficacy 15 Months after Vaccination

The efficacy of AL LiVac SRS following 15 months of immunization was assessed in a trial at VESO Vikan, where groups of Atlantic salmon had been vaccinated either with AJ LiVac SRS alone or in combination with AJ 5-1. This combination is widely used in the Chilean market. Groups vaccinated with AJ 5-1 3 weeks before AJ LiVac SRS and groups co-injected at the same time were included. A final group also tested the effect of boosting with AJ LiVac SRS 32 weeks after the initial AJ LiVac SRS vaccination. Vaccination was administered when the fish had an average weight of 56 g. The average weight at termination was 1399 g, 1496 g, and 1221 g in tanks 3, 5, and L23, respectively. Table 1 summarizes all vaccine groups.

The groups were mixed together in one holding tank after vaccination and held at 12 °C. A smoltification process was initiated two weeks after vaccination by exposing the fish to 24 h of daylight for 6 weeks and subsequent transfer to 25 ppt salinity and 12:12 light:dark signal. In order to simulate a winter and a summer scenario that is encountered with normal field use, the fish were divided into two tanks 13 months post-vaccination (60 days prior to challenge), and the temperature in one of the tanks was adjusted to 8 °C. Following these 60 days at 8 °C or 12 °C, both tanks were adjusted to 15 °C for challenge. Before challenge, ca. 5 fish from each of the groups vaccinated with AJ LiVac SRS alone, AJ LiVac SRS co-injected with AJ 5-1, or unvaccinated fish from both temperature regimes were transferred to a common tank. These fish were challenged by cohabiting the fish with 7 shedder fish that were intraperitoneally injected with *P. salmonis*. The remaining fish in the two original tanks were challenged by intraperitoneal injection. The cohabitation challenge was included in this study to test if protection found by intraperitoneal injection could also be replicated when challenge is performed through a natural route of infection.

The trial was part of a larger study that documented the duration of protection for Servicio Agricola y Ganadero (SAG), Chilean authorities, as part of the marketing authorization approval process for AJ LiVac SRS. The larger study contained challenges conducted on groups of fish from the same holding tank but with a shorter duration of immunization (3, 6, and 9 months). These results are not presented in this paper since the results were similar to those obtained with the longest duration (15 months).

### 2.5. Trial 2—Vaccine Efficacy at Different Immunization Temperatures

The effect of immunization temperature on vaccine protection was assessed in a trial at ILAB where fish vaccinated with AJ LiVac SRS were immunized at 8 °C, 10 °C, or 12 °C before intraperitoneal challenge with *P. salmonis*.

Atlantic salmon parrs (*n* = 420, Salmobreed, Stofnfiskur) kept in freshwater were acclimatized to either 8 °C, 10 °C or 12 °C in separate 160 L or 500 L tanks and a 12:12 light:dark regime. The fish were then starved for 24 h, anesthetized with Tricain (80 mg/L, PHARMAQ AS), and tagged either by shortening of maxillae and/or the Adipose fin or by injection of Elastomer VIE. While still under the same anesthetic period, the fish were vaccinated with AJ LiVac SRS or injected with sterile saline before they were transferred back to the holding tank.

Vaccinations at different temperatures were conducted at different dates to allow for the same number of degree days (ddg) to accumulate before a synchronized challenge of each temperature group. A total of 30 fish were individually weighed at each vaccination date to ensure that the population mean weight was comparable (mean weights 26 g, 26 g, and 27 g at the three dates, respectively). Details of vaccination and groups are given in Table 2.

Following a period of immunization, each group had accumulated approximately 450 ddg. The temperature was elevated as described above, and the groups were starved for 24 h, anesthetized using Tricain (80 mg/L, PHARMAQ AS), and injected intraperitoneally with 0.1 mL of a challenge inoculum containing 6 × 10^2^ TCID_50_ of *P. salmonis*. Following challenge, all groups were mixed into two parallel 500 L tanks containing freshwater at 15 °C.

### 2.6. Trials 3 and 4—Vaccination at Different Temperature Regimes and Sampling of Liver for Assessment of P. salmonis Growth In Vivo

Two related vaccination trials were conducted in order to link *P. salmonis* RNA levels in the liver after vaccination at different temperatures with observed vaccine efficacy.

In the first trial (Trial 3), Atlantic salmon were handled, vaccinated, and challenged essentially as described above but acclimatized in separate tanks (150 L, 500 L for the 10 °C group) to 7 °C, 8 °C, 10 °C, 11 °C and 15 °C before vaccination (Table 3). The mean weights of these groups at vaccination were 24 g, 23 g, 28 g, 24 g, and 24 g, respectively. One negative control group (*n* = 60) was injected with sterile saline and included in the tank acclimatized to 10 °C. The group vaccinated at 15 °C was kept at 15 °C only for 5 days before a 2 °C incremental daily decrease to 7 °C was carried out. The other groups were kept stable at their representative temperatures during immunization. Vaccination dates for each group were adjusted so that they would reach a similar number of degree days at challenge. At time points 1, 3-, 5-, 7- and 14 days post-vaccination, 6 fish per vaccinated group were euthanized by an overdose of Tricain (PHARMAQ AS), and the liver was sampled in RNAlater for subsequent RNA extraction and RT-PCR detection of *P. salmonis*. Challenge of the remaining fish was performed around 450 ddg (452–456 ddg) after vaccination when the temperature had been incrementally raised as described above to 15 °C and all the different temperature groups had been mixed into a common tank. Challenge was performed as described above by intraperitoneal injection of *P. salmonis*.

A second vaccination trial (Trial 4) was conducted to generate Ct values from fish that had been immunized at higher temperatures: 12 °C and 17 °C. Handling and vaccination procedures were carried out as described above on 30 fish per temperature. The fish were then sampled (*n* = 10 per sampling point) 3, 5, and 7 days after vaccination. Fish from this trial were not challenged.

The tanks used in Trials 3 and 4 also contained additional groups that were included in the trial for purposes other than the scope of this paper. They were, therefore, not sampled for PCR analysis. For these reasons, details from these groups are not reported here.

### 2.7. In Vitro Growth and Growth Curves

A pre-culture of the vaccine strain was prepared by inoculation of bacteria from a frozen stock into a flask containing 50 mL of a cell-free liquid medium. After cultivation for 4 days at 20 °C, the culture was spilt into new flasks containing 50 mL of a cell-free liquid medium by transferring 1 mL from the pre-culture. The flasks were incubated at 8 °C, 10 °C, 13 °C or 16 °C. Growth was measured using a Jenway 6310 Spectrophotometer to determine the OD600. The culture from the 8 °C incubator was sampled after 380 and 836 h and spread on Cysteine Heart Agar with 2% Blood (100 μL) and plates incubated for two weeks at 20 °C to verify viability *P. salmonis*.

### 2.8. RNA Extraction and Real-Time RT-PCR

The liver was sampled from freshly euthanized fish and transferred and stored in RNAlater. The samples were then sent to an accredited commercial laboratory for RNA extraction and Real-Time PCR analyses (PHARMAQ Analytiq, Bergen, Norway) using the PISCIsal-HI (16S ITS) and Atlantic salmon Elongation factor 1a assays. Raw Ct values from runs of 45 cycles were used in downstream analyses.

### 2.9. Data Handling and Statistics

Raw data were captured by handwriting into pre-made forms and then plotted into Microsoft Excel. Daily mortality logs were transformed into survival plot logs by Mort-O-Matic v0.2 [12] and imported into GraphPad Prism v9.4.1 (GraphPad Software, LLC) for statistical analysis. Ct values from Real-Time PCR analyses were obtained in Excel from PHARMAQ Analytiq (Bergen, Norway) and pasted directly into GraphPad Prism for downstream analyses. Negative samples were set to Ct = 45. Ct values from the PISCIsal-HI assay were also normalized against Ct values for Elongation factor 1a using the Pfaffl method [13] in a separate analysis to ensure that factors in the sampling or RNA processing did not affect the conclusion.

## 3. Results

### 3.1. Vaccination of Atlantic Salmon with AJ LiVac SRS Is Highly Protective against Mortality Caused by Subsequent Challenge with P. salmonis for at Least 15 Months

To evaluate the efficacy of AJ LiVac SRS, an experimental challenge by intraperitoneal injection was conducted 15 months after immunization (Trial 1). Results showed near full protection against *P. salmonis*-induced mortality in groups vaccinated with AJ LiVac SRS, both when held at a stable temperature of 12 °C and when subjected to a 60-day winter simulation at 8 °C during the immunization period (Figure 1). Vaccination with the combination of AJ 5-1 and AJ LiVac SRS affected the performance of AJ LiVac SRS negatively, but only when AJ 5-1 was given 3 weeks prior to AJ LiVac SRS administration (AJ5-1 (−3) + LiVac). Boosting this group with a second dose of AJ LiVac SRS 32 weeks after the first administration restored the same effect as found when administering AJ LiVac SRS alone. Challenges performed on fish that had been immunized for 3, 6, or 9 months are not included in this paper since the results were similar to those obtained 15 months after immunization.

Challenge by cohabitation 15 months after immunization led to mortality in all control fish starting 40 days after exposure to shedder fish. All fish vaccinated either with AJ LiVac SRS alone or with the AJ LiVac SRS + AJ 5-1 co-injection survived, with the exception of one fish that died as a result of jumping out of the tank 4 days after challenge (Figure 2).

### 3.2. Vaccine Efficacy Is Dependent on Immunization Temperature

The effect of AJ LiVac SRS was further tested in the intraperitoneal challenge model following different temperature regimes during immunization (Trial 2). Mortalities due to SRS were observed in the negative control groups starting 16 days after challenge, with all fish dying within 22 days post-challenge in both parallel tanks (Figure 3). Significant mortalities also accumulated for the AJ LiVac SRS vaccinated groups that had been immunized at 8 °C, reaching 32–66% accumulated mortality at study termination. No mortality was observed in the groups immunized at 10 °C and 12 °C, with the exception of one dead fish immunized with batch 15298 at 12 °C.

### 3.3. Lack of In Vitro Bacterial Growth at Low Temperature

Growth curves for the AL20542 vaccine strain were generated based on the optical density of the growth medium over time in order to test the temperature sensitivity of the bacterium. The measured data were fitted to curves with an exponential shape followed by a plateau phase. A high amount of the variation in the dataset could be explained by this curve fit for temperatures 16 °C, 13 °C, and 10 °C (r_2_ = 0.92, 0.86, and 0.80, respectively), suggesting efficient bacterial growth at these temperatures. The OD_600_ measured at the plateau phase increased with temperature. No such growth was detected for bacteria grown at 8 °C (Figure 4).

### 3.4. Temperature-Dependent Loss of Protection Is Associated with Reduced In Vivo Growth the First 5 Days Post-Vaccination

Fish were vaccinated with AJ LiVac SRS at different temperatures ranging from 7 °C to 17 °C (Trials 3 and 4). *P. salmonis* RNA was detected by RT-PCR in the liver of most sampled fish after vaccination. Mean Ct values correlated negatively with temperature during immunization; thus, the amount of *P. salmonis* RNA in the liver increased with the temperature (Figure 5A). This correlation was found to be statistically significant for samplings made 5 and 7 days after vaccination (Spearman correlation, *p* = 0.113 d.p.v., *p* = 0.0075 d.p.v. and *p* = 0.017 d.p.v.) Normalization of Ct values against the internal reference gene Elongation factor 1a did not considerably change this pattern (Appendix A).

A strong protection against experimental challenge with *P. salmonis* was observed for groups vaccinated at 10 °C, 11 °C and 15 °C, while only limited protection was observed in the two groups vaccinated at 7 °C and 8 °C. The group that was vaccinated at 15 °C and then transferred to 7 °C after 5 days had retained its protection, demonstrating that 5 days of immunization at high temperature is sufficient to build a significant immune response after vaccination with AJ LiVac SRS (Figure 6).

## 4. Discussion

The live attenuated vaccine, ALPHA JECT LiVac^®^ SRS, has been shown in laboratory challenge models to be effective in reducing mortalities caused by *P. salmonis* for a prolonged period, covering the approximate lifespan of farmed Atlantic salmon. However, our study has also revealed that the vaccine’s efficacy can be compromised by improper use. More specifically, we found that the amount of *P. salmonis* RNA in the liver after vaccination (as a proxy for vaccine propagation *in vivo*) is temperature-dependent, and immunization at 8 °C or below results in a significant loss of efficacy. Our findings are supported by the *in vitro* growth of the vaccine strain AL20542, which appears to be completely stalled at these temperatures. Active propagation of the vaccine strain is likely to be a prerequisite for exploiting the full potential of live attenuated vaccines. These results therefore identify an important pitfall in the use of AJ LiVac SRS and could provide an explanation for some of the recent controversy around the efficacy of this vaccine [8,9]. Surveillance programs and scientific projects that aim at optimizing the use of this and other live vaccines for poikilothermic animals must take this factor into account and preferably utilize tools that can verify that the animals have indeed been exposed to a propagating vaccine organism.

While live vaccines constitute the oldest known vaccine technology and are widely regarded as a robust way of inducing immunity in warm-blooded animals [14], their use in poikilothermic animals has been limited. Fish constitute the main group of poikilothermic animals where vaccination is extensively carried out, and only a few examples of live fish vaccines have reached the stage of commercialization [15]. Temperature is clearly a factor that needs addressing for the safe and efficacious use of live vaccines in such hosts since it is an impactful regulator of most biological processes. The speed and level at which the fish immune system is activated is often thought to increase with temperature [16]. However, perhaps more important for the efficacy of a live vaccine is the ability of the vaccine organism to propagate [17,18]. Propagation *in vitro* of the AJ LiVac SRS vaccine-strain AL20542 slowed gradually down with lower temperature and appeared to stop completely at 8 °C. A similar trend was found *in vivo* by correlating Ct values in the liver with temperature. The continuous nature of this process suggests that the effect of vaccination is not likely to be a binary phenomenon that is turned on at a certain threshold temperature, but rather a process where the effect increases gradually with temperature.

Liver samples were positive with high Ct values in groups immunized at lower temperatures at the earlier sampling points, suggesting some presence of vaccine-strain RNA. This RNA could, however, originate from intact, non-propagating bacterial cells covering the surface of the liver, as would be expected after an intraperitoneal injection. Vaccination under such conditions is therefore likely to elicit an immune response similar to an inactivated antigen, which will be quite limited, particularly given the low antigenic dose and lack of an effective adjuvant in AJ LiVac SRS [19].

Controversies regarding vaccine efficacy can evolve into polarized public opinions. Vaccines against intracellular pathogens offer relief of mortality, disease symptoms, and pathogen production and spread, but often not sterile immunity [20,21]. The sustained presence of such diseases after the introduction of a vaccine is therefore common, and not evidence of lack of protection. Moreover, the estimation of the true field efficacy of these vaccines purely based on production data and official diagnoses is challenging. One problem can be the lack of dissection of SRS cases into different vaccines and vaccination regimes [22]. Another issue is the handling of an SRS diagnosis in the Chilean industry, which often results in immediate treatment with antibiotics to secure a high probability of treatment success [6], not leaving much window for true efficacy evaluation. Laboratory challenge models are therefore often used to evaluate vaccine performance. These will, however, never fully emulate field conditions. Even the cohabitation model that mimics natural transmission will only serve to rank vaccines, vaccination regimes, and practices in the strictest sense. It is therefore a fallacy to assume that any quantitative metric of protection from laboratory trials can be directly transferred to field protection. This can be exemplified by the complete protection measured 15 months post-vaccination in our cohabitation model despite continuous SRS outbreaks in the field, which has also been observed for other fish vaccines [21]. Laboratory models can, however, be very helpful for working toward the optimal use of a vaccine by ranking different strategies against each other. Administration of a water-in-oil-based multivalent vaccine (AJ 5-1) three weeks before AJ LiVac SRS administration appeared, for example, to reduce the protective effect of AJ LiVac SRS (Figure 1). While we have not studied the cause of this interference, it is likely to be associated either with the local unspecific immune response in the intraperitoneal cavity following AJ 5-1 vaccination or by a specific immunity towards *P. salmonis* provided by this vaccine. Both these factors could potentially interfere with the ability of the AJ LiVac SRS vaccine strain to propagate the days after vaccination.

In addition to temperature during immunization and interference with other vaccines presented here, other researchers have produced data in challenge models to suggest that genetic breed may be of some importance for protection [23]. The presence of sea lice worsens the clinical picture during SRS and would clearly be beneficial to reduce [24]. It has also been suggested that the protection provided by AJ LiVac SRS may not capture the serological diversity of *P. salmonis* strains that cause disease in Chile [9]. The Chilean disease situation appears to be dominated by two distinct groups of *P. salmonis*—EM90-like and LF89-like strains [25]—where AJ LiVac SRS is based on an EM90-like strain. Although these strains show some phenotypic differences [11,25], the implications of these differences for vaccine efficacy still remain to be elucidated.

SRS remains a problem in Chilean fish farming. The use of efficacious vaccines is probably not a complete solution to the problem, but it still offers an important countermeasure to relieve the burden of disease. It is, however, important that the vaccine is used optimally. Live attenuated vaccines, such as AJ LiVac SRS, can be sensitive to erroneous use, as identified here. Fish producers and other users of the vaccine need to be cautious so that fish populations are not vaccinated under conditions that fail to induce proper immunity. The temperature during immunization is one very crucial factor to control in that regard.

## Figures and Tables

**Figure 1 vaccines-12-00416-f001:**
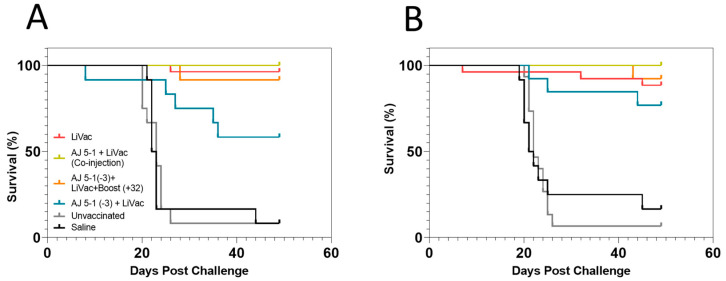
Survival plots of groups vaccinated with AJ LiVac SRS following intraperitoneal challenge with *P. salmonis*. Fish were vaccinated with AJ LiVac SRS alone or combinations of AJ 5-1 and AJ LiVac SRS as indicated and then allowed to develop immunity for 15 months prior to the challenge. One tank was immunized at 12 °C for the whole 15-month period (**A**), while the other tank was held at 12 °C and then given a 60-day “winter” at 8 °C (**B**) prior to the challenge.

**Figure 2 vaccines-12-00416-f002:**
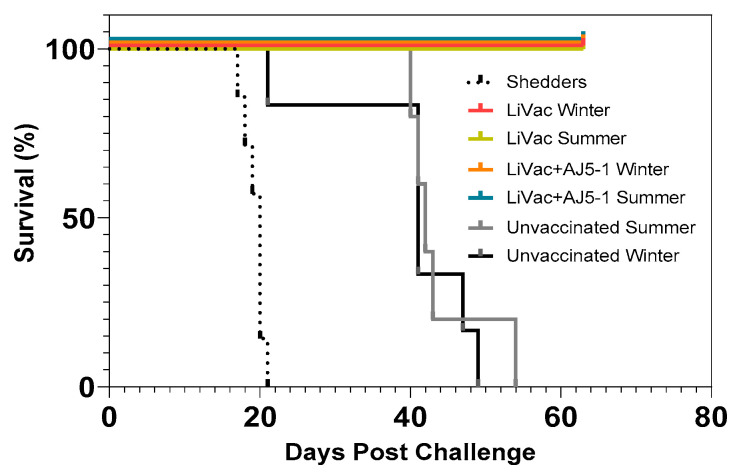
Survival plot of groups vaccinated with AJ LiVac SRS after 15 months of immunization and subsequent cohabitation challenge with *P. salmonis*. Fish were vaccinated with the indicated vaccination regimes and allowed to develop immunity for 15 months. The fish were sourced from the same immunization groups as those in Figure 1, where “Summer” indicates 12 °C for the full 15 months, while “Winter” indicates 12 °C except for a simulated winter of 8 °C for 60 days prior to challenge. All groups were challenged in the same tank at 15 °C. One fish that died in the group vaccinated with AJ LiVac SRS + AJ 5-1 (8 °C) was omitted from this study since the cause of death was jumping out of the tank.

**Figure 3 vaccines-12-00416-f003:**
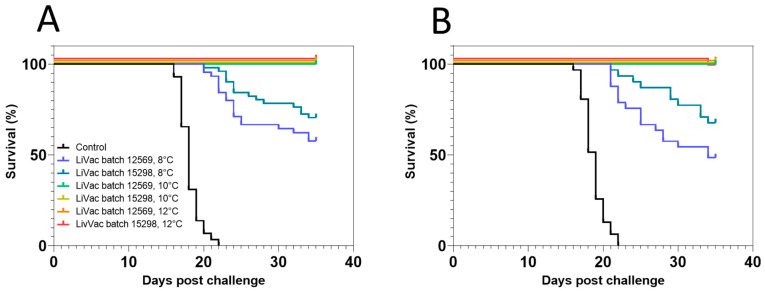
Survival plots of groups vaccinated with AJ LiVac SRS and immunized at different temperatures followed by subsequent intraperitoneal challenge with *P. salmonis*. Fish were vaccinated with two different commercial batches of AJ LiVac SRS (12569 and 15298) in two parallel tanks (**A**,**B**). Immunization time was adjusted to temperature so that each group had accumulated the same amount of degrees days after vaccination on the day of challenge (ca. 450 ddg).

**Figure 4 vaccines-12-00416-f004:**
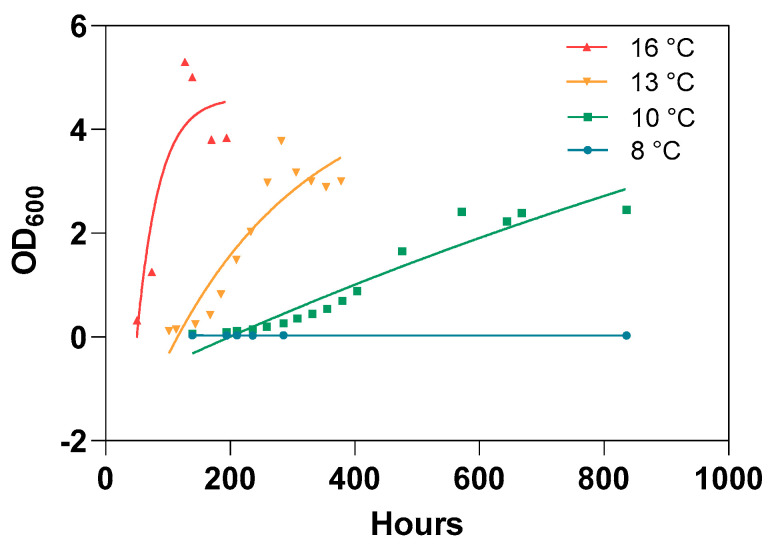
In vitro growth curves for the vaccine strain AL20542. The vaccine-strain AL20542 was grown in a 50 mL cell-free liquid medium in ventilated shaker flasks at the indicated temperatures. The OD_600_ of the medium was assessed at the indicated time points. Lines indicate an exponential growth function with a plateau phase fitted to the data.

**Figure 5 vaccines-12-00416-f005:**
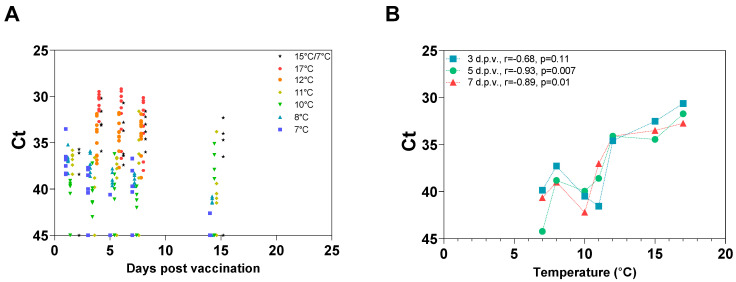
Ct values in the liver following vaccination with AJ LiVac SRS at different temperatures. (**A**) Ct values in livers from individual fish. The fish were vaccinated and held at different temperatures, as indicated in the figure, and livers were samples at the indicated time points from 6 or 10 fish per group. (**B**) Plotting of mean Ct values from each temperature group at sampling points 3-, 5- and 7-days post-vaccination. Sampling days 1 and 14 were not analyzed since no data from the 12 °C and 17 °C groups were available at these time points. A dotted line was used to connect data points from the same sampling day. A Spearman correlation between temperature and mean Ct value was computed, and Spearman’s r- and *p*-values were indicated for each sampling day.

**Figure 6 vaccines-12-00416-f006:**
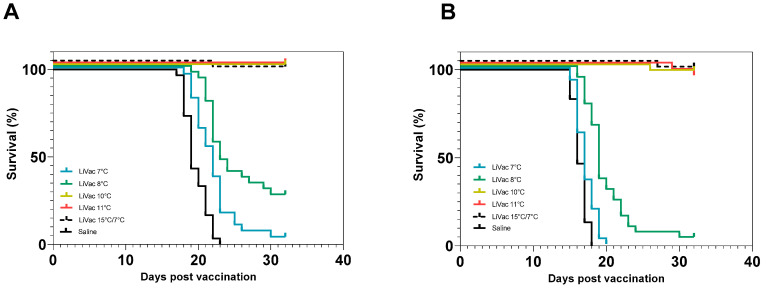
Survival plots of the groups from Trial 3 were vaccinated with AJ LiVac SRS and immunized at different temperatures, followed by subsequent intraperitoneal challenges in parallel tanks (**A**,**B**) with *P. salmonis*. The group LiVac 15 °C/7 °C was vaccinated and then held at 15 °C for 5 days before lowering temperature to 7 °C. Ct values for *P. salmonis* the first 14 days after vaccination for these groups are shown in Figure 5.

**Table 1 vaccines-12-00416-t001:** Challenge with *P. salmonis* of fish immunized for 15 months with vaccine combinations containing AJ LiVac SRS (Trial 1). The vaccines AJ 5-1 and AJ LiVac SRS were administered either with a 3-week lag (−3 w) or at the same time where indicated. One group was also boosted (Boost) 32 weeks after the initial AJ LiVac SRS administration. All vaccinations were conducted at 12 °C, and fish were then held at this temperature the whole immunization period (tank 5) or given a simulated winter with 60 days at 8 °C (tank 3) before elevation to challenge temperature (15 °C). The vaccine groups challenged in tank L23 were sourced from tanks 3 and 5 before challenge. AF = Adipose fin, RM = Right maxillae, LM = Left maxillae, PIT = PIT-tag.

Tank	Vaccine Group	Tag	N	Winter	Challenge By
3	AJ5-1(−3 w)—LiVac	AF	12	Yes	Intraperitoneal injection
	AJ5-1 + LiVac	RM	19	Yes	
	LiVac	LM	27	Yes	
	Saline	RM + AF	12	Yes	
	AJ5-1(−3 w)—LiVac—LiVac (Boost)	LM + AF	12	Yes	
	Unvaccinated	PIT	12	Yes	
5	AJ5-1(−3 w)—LiVac	AF	13	No	Intraperitoneal injection
	AJ5-1 + LiVac	RM	18	No	
	LiVac	LM	26	No	
	Saline	RM + AF	12	No	
	AJ5-1(−3 w)—LiVac—LiVac (Boost)	LM + AF	13	No	
	Unvaccinated	PIT	15	No	
L23	AJ5-1 + LiVac	RM	5	No	Cohabitation
	LiVac	LM	5	No	
	Unvaccinated	PIT	5	No	
	AJ5-1 + LiVac	RM + AF	4	Yes	
	LiVac	LM + AF	5	Yes	
	Unvaccinated	PIT + AF	6	Yes	
	Shedders	None	7	No	Intraperitoneal injection

**Table 2 vaccines-12-00416-t002:** Vaccination with AJ LiVac SRS at different temperatures (Trial 2). The groups were immunized at different temperatures and later mixed together in two parallel tanks for challenge with *P. salmonis*. The groups were vaccinated at different temperatures so that they would accumulate the same amount of degree days at challenge. AF = Adipose fin, RM = Right maxillae, LM = Left maxillae, VIE = Elastomer VIE.

Group	Vaccine	N	Tag	Tanks	Immunization Temperature	Date of Vaccination
L1-8	LiVac (12569)	30 + 30	RM	C6T1 * + C6T2 *	8 °C	13 February 2019
L2-8	LiVac (15298)	30 + 30	LM	C6T1 * + C6T2 *		
Control-8	0.9% NaCl	30 + 30	AF	C6T1 * + C6T2 *		
L1-10	LiVac (12569)	30 + 30	RM + AF	C3T1 + C3T2	10 °C	25 February 2019
L2-10	LiVac (15298)	30 + 30	LM + AF	C3T1 + C3T2		
L1-12	LiVac (12569)	30 + 30	VIE Red	C3T3 + C3T4	12 °C	4 March 2019
L2-12	LiVac (15298)	30 + 30	VIE Green	C3T3 + C3T4		

* Tanks C6T1 and C6T2 were 500 L. The remaining tanks were 160 L.

**Table 3 vaccines-12-00416-t003:** Vaccination at different temperatures was done to correlate temperature with liver Ct values and protection (Trials 3 and 4). The groups were kept in separate tanks for the different temperatures during immunization. Groups from Trial 4 were not challenged. RM = Right maxillae, AF = Adipose fin, VIE = Elastomer VIE.

Trial	Vaccine Group	Immunization Temperature	Tag	N for Challenge	N for Liver Sampling	Sampling Days p.v.	Date for Vaccination
Trail 3	LiVac	7 °C	RM	60	30	1, 3, 5, 7,1 4	3 March 2016
	LiVac	8 °C	AF	60	30	1, 3, 5, 7,1 4	29 March 2016
	LiVac *	10 °C	RM + AF	60	30	1, 3, 5, 7,1 4	8 April 2016
	LiVac	11 °C	VIE Blue + RM	60	30	1, 3, 5, 7,1 4	15 April 2016
	LiVac	15 °C /7 °C	VIE Red + RM	60	30	1, 3, 5, 7,1 4	31 March 2016
	0.9% NaCl *	10 °C	VIE Green	60	0	-	8 April 2016
Trail 4	LiVac	12 °C	-	0	30	3, 5, 7	11 March 2016
	LiVac	17 °C	-	0	30	3, 5, 7	11 March 2016

* Kept in the same tank during immunization.

## Data Availability

The raw data supporting the conclusions of this article will be made available by the authors on request.

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
