# Peer review of "The Effect of an Attenuated Live Vaccine against Salmonid Rickettsial Septicemia in Atlantic Salmon (Salmo salar) Is Highly Dependent on Water Temperature during Immunization"

_vaccines, 2024, doi:10.3390/vaccines12040416_

Round 1

Reviewer 1 Report

Comments and Suggestions for Authors

The authors Olsen et al demonstrated that the immunogenicity and protective efficacy of a commercially available live vaccine against rickettsial septicaemia in Atlantic Salmon is regulated by the temperature during the vaccination period. The live attenuated strain needs a certain temperature to propagate in the fish for giving protection against bacterial challenges. This research is very important for the fish industries where live vaccines are used preferably than antibiotics to protect the fish from diseases and help to reduce the chance of getting multi drug resistant bacteria. The manuscript is well designed, and the result are presented adequately. Some Specific comments are as below:

1. In methods 2.2, in the last line, briefly discuss the process of thawing and dilution of the AJLiVac.

2. Line 95: Spell TCID.

3. Figure 1A: Discuss in detail about the less effect of AJ5-1 (-3w) LiVac group in discussion section. Improve the quality of the text in Figure 1AB and 3AB.

4. There are two figures with the name of Figure 5. Either name the latter figure as Figure 6 or mark them as 5C and 5D. Change the result text accordingly.

Author Response

The authors Olsen et al demonstrated that the immunogenicity and protective efficacy of a commercially available live vaccine against rickettsial septicaemia in Atlantic Salmon is regulated by the temperature during the vaccination period. The live attenuated strain needs a certain temperature to propagate in the fish for giving protection against bacterial challenges. This research is very important for the fish industries where live vaccines are used preferably than antibiotics to protect the fish from diseases and help to reduce the chance of getting multi drug resistant bacteria. The manuscript is well designed, and the result are presented adequately. Some Specific comments are as below:

  1. In methods 2.2, in the last line, briefly discuss the process of thawing and dilution of the AJLiVac.

Ok, we have expanded on the text describing the method of preparing the vaccine.

  1. Line 95: Spell TCID.

Ok, done.

  1. Figure 1A: Discuss in detail about the less effect of AJ5-1 (-3w) LiVac group in discussion section. Improve the quality of the text in Figure 1AB and 3AB.

Ok. We have added a section in the discussion to address this issue.

The figure-texts in Figs 1 and 3 have also been expanded on and improved.

  1. There are two figures with the name of Figure 5. Either name the latter figure as Figure 6 or mark them as 5C and 5D. Change the result text accordingly.

Thank you for pointing this out. This has been changed to Figure 6.

Reviewer 2 Report

Comments and Suggestions for Authors

From the perspective of temperature, the authors verified the effect of temperature on the effect of vaccination after vaccination. This trial design idea is very good. However, the scientific validity of the paper's data is debatable. The specific comments are as follows:

1. The title seems to need to be revised. Because the authors conclude that five days after vaccination, not the first day, the temperature can greatly affect the effectiveness of the vaccine. Therefore, the title is proposed to be changed to: Effect of an attenuated live vaccine against Salmonid Rickettsial Septicaemia in Atlantic salmon (Salmo salar) is highly dependent on temperature the first five days after vaccination.

2. Regarding the results of Part 3.4, the reviewers questioned the scientific validity of these results.

It is unscientific to use Ct values to represent the number of bacterial nucleic acids in the fish.

In addition, Real Time RT-PCR can measure the number of nucleic acids of P. salmonis, including nucleic acids in live P. salmonis and dead P. salmonis, so it is not scientific to reflect the effectiveness of vaccines only from the number of bacterial nucleic acids.

Comments on the Quality of English Language

Minor editing of English language required.

Author Response

  1. The title seems to need to be revised. Because the authors conclude that five days after vaccination, not the first day, the temperature can greatly affect the effectiveness of the vaccine. Therefore, the title is proposed to be changed to: Effect of an attenuated live vaccine against Salmonid Rickettsial Septicaemia in Atlantic salmon (Salmo salar) is highly dependent on temperature the first five days after vaccination.

  1. We have purposely been reluctant to state “five days” explicitly in the title. The reason for this is that we believe that this can give an overly strong impression that five days is optimal under all conditions. We have not explored this time-dependency sufficiently in this manuscript to defend such a statement, and we expect that it is temperature-dependent so that more than five days could be necessary for colder temperatures. We have now suggested a new title to be more general with regard to the timing of this effect.

  1. Regarding the results of Part 3.4, the reviewers questioned the scientific validity of these results.

It is unscientific to use Ct values to represent the number of bacterial nucleic acids in the fish.

In addition, Real Time RT-PCR can measure the number of nucleic acids of P. salmonis, including nucleic acids in live P. salmonis and dead P. salmonis, so it is not scientific to reflect the effectiveness of vaccines only from the number of bacterial nucleic acids.

Comments on the Quality of English Language Minor editing of English language required.

  1. We have conducted an additional analysis to normalize Ct-values according to the Pfaffl-method against an internal housekeeping gene, elongation factor 1a. The results are provided as supplementary material to the manuscript and shows that the pattern of correlation between bacterial RNA copies and temperature remains both when using raw Ct-values and normalized Ct-values.

We agree with the general reflections of the reviewer regarding limitations of real-time PCR, however, we have not concluded on effectiveness of vaccination based on PCR-results. Nor have we concluded on the absolute in vivo viability of the vaccine strain in these temperature groups. Contrary, we have specified in the discussion that some of the detected RNA probably comes from dead or non-propagating cells, thus highlighting the limitation of this method.

The PCR-results are important to include because the differences in protection (measured by experimental challenge) correlate with the strength of the PCR-signal early after vaccination and with temperature. This correlation supports the notion that temperature affects both in vivo vaccine-strain activity and subsequent protection. Moreover, the finding of in vitro growth curves being highly affected in the same temperature interval further identifies lack of bacterial activity/propagation to be the main mechanism behind reduced effect at lower temperatures.

We have tried to amend the text in the discussion to be more clear on this distinction and the limitations of PCR.

In addition to these revisions we have also made the following changes:

  1. The naming of the vaccines throughout the manuscript has been made more consistent, and hopefully more easy to read.
  2. Minor changes to improve readability in several places of the manuscript.
  3. The statistical analysis based on the data in Figure 5b has been changed from a linear regression to a Spearman correlation. The reason for this is that the linearity of the data can be questioned. The Spearman correlation does not require linearity, and we therefore avoid that uncertainty. For this reason the figure was also slightly revised: regression lines were removed and simple connecting lines were used to make it easier for the readers to follow each group/day.

Round 2

Reviewer 2 Report

Comments and Suggestions for Authors

The manuscript can be accepted.